# An Update on Implant-Associated Malignancies and Their Biocompatibility

**DOI:** 10.3390/ijms25094653

**Published:** 2024-04-24

**Authors:** Grace C. Keane Tahmaseb, Alexandra M. Keane, Jose A. Foppiani, Terence M. Myckatyn

**Affiliations:** 1Division of Plastic and Reconstructive Surgery, Washington University School of Medicine, St. Louis, MO 63130, USA; gkeane@wustl.edu (G.C.K.T.); amkeane@wustl.edu (A.M.K.); 2Division of Plastic Surgery, Beth Israel Deaconess Medical Center, Harvard Medical School, Boston, MA 02215, USA; jfoppian@bidmc.harvard.edu

**Keywords:** implant-associated malignancies, BIA-ALCL, breast implants, orthopedic implants

## Abstract

Implanted medical devices are widely used across various medical specialties for numerous applications, ranging from cardiovascular supports to orthopedic prostheses and cosmetic enhancements. However, recent observations have raised concerns about the potential of these implants to induce malignancies in the tissues surrounding them. There have been several case reports documenting the occurrence of cancers adjacent to these devices, prompting a closer examination of their safety. This review delves into the epidemiology, clinical presentations, pathological findings, and hypothesized mechanisms of carcinogenesis related to implanted devices. It also explores how the surgical domain and the intrinsic properties and biocompatibility of the implants might influence the development of these rare but serious malignancies. Understanding these associations is crucial for assessing the risks associated with the use of medical implants, and for developing strategies to mitigate potential adverse outcomes.

## 1. Introduction

Implantable medical devices are utilized in nearly all surgical specialties to treat a myriad of conditions. According to the American Medical Association, 10% of all people will have a medical device implanted during their lifetime [1]. These devices include orthopedic artificial joints, cardiac pacemakers, intravascular stents, breast implants, dental implants, and indwelling catheters. Implanted medical devices are usually constructed from metals, ceramics, polymers, or a combination thereof.

Though these devices offer patients extended longevity and quality of life, they are not without risk, with infection, device failure, and compromise of surrounding tissues among the most common complications. In recent years, the possibility of implant-associated malignancies has gained traction, driven in part by the recent commentary on breast implant-associated malignancies.

However, implant-associated malignancies are not limited to breast implants. Over the last 50 years, several surgical fields have published case reports on malignancies related to implanted foreign bodies. In this review, we summarize the current knowledge on implant-related malignancies as relates to their surgical field, pathology, and proposed molecular underpinnings. We examine the properties of the materials used to create these devices and modifications intended to enhance their biocompatibility.

## 2. Methodology

### 2.1. Study Design and Search Strategy

This literature review utilizes case reports, clinical trials, other reviews, and cohort studies from the last fifty years to investigate the incidence and characteristics of malignancies associated with implantable medical devices, and to highlight the current evidence for the role of implant materials in carcinogenesis. It also aims to highlight the most common materials used in implants, along with their properties and biocompatibility. Search strategies were developed and conducted in partnership with a librarian to ensure comprehensive coverage of the topic, utilizing appropriate search terms for each section. The search was carried out across several databases, including PubMed, Scopus, and Web of Science [2,3,4,5,6].

### 2.2. Inclusion and Exclusion Criteria with Material Analysis

Studies were selected that reported malignancies directly associated with implantable medical devices, providing explicit information about the implant material, the duration of implantation, and the subsequent development of malignancy. Emphasis was placed on research detailing these materials’ biocompatibility, chemical properties, release of metal ions, wear debris, surface modifications, and interactions with biological environments, so as to understand their possible roles in promoting malignant transformations.

Excluded from the review were case reports lacking detailed documentation of implant materials, as well as studies concentrating solely on the mechanical failures of implants, infections, or other complications. 

### 2.3. Data Synthesis

A narrative synthesis was conducted to amalgamate the collected data, focusing on linking implant materials to the development of malignancies. This method allowed for a thorough exploration of the complex interplay between various implant materials and cancer, highlighting specific materials implicated in malignancies and examining their interactions with biological tissues.

## 3. Implants’ Materials and Biocompatibility

### 3.1. Titanium

#### 3.1.1. Mechanical Properties

Commercially pure titanium (CpTi) grades and titanium alloys, such as Ti-6Al-4V, display varying degrees of strength, ductility, and corrosion resistance, attributes that are directly influenced by the oxygen content and the incorporation of alloying elements [7,8,9,10]. Grade IV CpTi, notable for its high oxygen content, showcases superior mechanical strength, underscoring the critical role of molecular composition in determining an implant’s mechanical properties [7,8,9,10]. Alloying elements can be further categorized into alpha (α) stabilizers, like aluminum, and beta (β) stabilizers, such as vanadium, each affecting the α and β phases of the titanium’s microstructure, thereby modulating its overall strength and corrosion resistance [7,8,9,10]. Wu et al. (2023) highlight the importance of specific mechanical properties—hardness, tensile strength, yield strength, and fatigue strength—in contributing to the long-term stability and functionality of oral implants [7,8,9,10]. The elastic modulus, in particular, is essential for ensuring that stress transfer to the surrounding bone remains within physiological limits, preventing bone resorption and fracture [7,8,9,10]. This detailed analysis accentuates the necessity of aligning the elastic modulus of implants with that of human bone to optimize biomechanical compatibility and facilitate successful osseointegration, bridging the gap between molecular structure and biomechanical functionality.

#### 3.1.2. Surface Characteristics and Bioactivity

The molecular configuration of titanium surfaces, which are primarily composed of titanium dioxide (TiO_2_), plays an indispensable role in the material’s interaction with biological environments, significantly influencing both its bioactivity and corrosion resistance [8,11,12]. The high affinity of titanium for oxygen results in the spontaneous formation of a TiO_2_ layer upon exposure to air or water, a characteristic that facilitates the metal’s exceptional corrosion resistance and bioactivity [8,11,12]. This oxide layer can crystallize into several forms—rutile, anatase, and brookite—each possessing distinct molecular structures that crucially affect the material’s biological compatibility [8,11,12]. Specifically, the anatase form of TiO_2_ is noted for its potential to enhance osseointegration, a pivotal factor in the success of dental implants [8,11,12]. Furthermore, advancements in surface engineering have enabled the creation of nanostructured modifications such as nanotubes and nanopores on these titanium surfaces [8,11,12]. Such nanostructures, developed through techniques like anodization, significantly impact cellular responses and osseointegration by mimicking the bone’s natural architecture [8,11,12]. This biomimetic approach promotes cellular adhesion and proliferation, thereby facilitating the integration of titanium-based implants with the surrounding biological tissue.

#### 3.1.3. Nanoscale Modifications and Biocompatibility

Nanostructured modifications on titanium surfaces, such as nanotubes, nanoparticles, and nanopores, have been identified as critical enhancers of biocompatibility and bioactivity, closely mimicking the natural bone structure at the molecular level and significantly improving implant performance [8,11,13,14]. These nanostructures, created through advanced techniques like anodization, not only replicate the bone’s nanoscale architecture but also significantly impact cell–material interactions by promoting cell adhesion, proliferation, and differentiation [8,11,13,14]. Silva et al. (2022) and Yan et al. (2022) both underscore the importance of such modifications in fostering positive cellular responses and accelerating osseointegration, with surface wettability altered by nanostructures playing a pivotal role in protein adsorption and cell attachment [8,11,13,14]. Further expanding on this, Chen et al. (2013) delved into the methods for immobilizing bioactive molecules on titanium surfaces, including physical adsorption, chemically covalent bonding, and biomimetic incorporation, to control tissue–implant interactions at a molecular level, thereby inducing cellular responses that enhance bone formation and integration [8,11,13,14]. Kumar et al. (2019) further explored the creation of biomimetic surfaces through nanoscale topography modifications, employing techniques such as titanium plasma spraying, grit blasting, acid etching, and anodic oxidation [8,11,13,14]. These approaches aim to mimic the extracellular matrix, significantly improving the capacity of titanium implants to anchor into bone and enhance molecular interactions at the implant interface, illustrating a concerted effort to leverage nanostructured modifications for optimizing the functionality and integration of titanium-based implants.

### 3.2. Cobalt–Chromium–Molybdenum

#### 3.2.1. Mechanical Properties

Cobalt–chromium (Co-Cr) alloys stand out in the medical field for their exceptional mechanical properties, essential for a wide range of implant technologies [15,16,17,18,19,20] These alloys are particularly valued for their high strength, temperature endurance, and wear resistance, critical for the success of orthopedic, dental, and cardiovascular implants. In spinal surgeries, Co-Cr alloy wires demonstrate superior tensile strength, significantly enhancing the correction of deformities without the risk of failures or complications [15,16,17,18,19,20]. In dental applications, Co-Cr alloys processed through CAD/CAM techniques exhibit optimized microstructures and mechanical properties, crucial for the longevity and success of dental implants [15,16,17,18,19,20]. Moreover, the use of selective laser melting (SLM) followed by heat treatment significantly improves the ductility and retentive forces of Co-Cr alloys used in removable partial denture frameworks, offering an advanced fabrication strategy [15,16,17,18,19,20]. The optimization of Co-Cr alloy stents for coronary interventions also highlights the alloy’s adaptability, with improved axial retraction performance and flexibility being crucial to enhancing stent functionality [15,16,17,18,19,20]. Collectively, these properties underscore the versatility and reliability of Co-Cr alloys across various medical applications, proving them indispensable in the advancement of implant technologies.

#### 3.2.2. Surface Characteristics and Bioactivity

Surface characteristics of cobalt–chromium (Co-Cr) implants significantly influence their performance, osseointegration, and longevity in orthopedic and dental applications [20,21,22,23]. Cobalt–chromium alloys are known for their high strength, temperature endurance, and wear resistance, making them ideal for such applications [20,21,22,23]. The success of these implants is closely related to their ability to achieve direct bone-to-implant contact without the interposition of non-bone tissue, a process known as osseointegration [20,21,22,23]. However, the degree of bone formation around the implant can be affected by the implant’s surface characteristics, including its topography and chemical composition [20,21,22,23]. Porous coatings on Co-Cr implants, for example, can promote bone ingrowth, enhancing stability and long-term success [20,21,22,23]. Surface roughness and hydrophobicity play crucial roles in determining the interaction between the implant and the biological environment [20,21,22,23]. Studies comparing Co-Cr to titanium implants have found differences in surface roughness levels, but more notably, Co-Cr’s surface properties contribute to its hydrophobic nature, potentially reducing bacterial adhesion compared to more hydrophilic surfaces like titanium [20,21,22,23]. This characteristic is beneficial for minimizing plaque accumulation and subsequent infection risks [20,21,22,23]. Surface modifications, such as nitrogen ion implantation, have been explored to improve the surface hardness and frictional properties of Co-Cr implants [20,21,22,23]. These modifications aim to enhance the wear resistance and longevity of the implants. However, the benefits of increased surface hardness and reduced friction might decrease over time when the implant is in the body, indicating a dynamic interaction between surface modifications and biological processes.

#### 3.2.3. Nanoscale Modifications and Biocompatibility

Research on cobalt–chromium–molybdenum (CoCrMo) implants has focused on enhancing their integration and functionality through nanoscale surface modifications [23,24,25]. Studies have shown that surface modifications can significantly impact the osteogenic differentiation potential of stem cells and reduce biofilm formation by pathogens like Staphylococcus aureus and Staphylococcus epidermidis, crucial for preventing post-implantation infections [23,24,25]. Furthermore, advancements in additive manufacturing have allowed for the development of calcium phosphate-reinforced CoCrMo alloys, showing promising biotribological performance and biocompatibility for load-bearing implants [23,24,25]. These modifications are essential for improving wear resistance and osseointegration, ensuring the longevity and success of implants. This collective research underscores the critical role of surface modifications in advancing CoCrMo implant technology.

### 3.3. Stainless Steel

#### 3.3.1. Mechanical Properties [26,27,28]

Stainless steel—notably, the 316L variant—is widely chosen for bone surgery and internal fixation devices due to its impressive combination of mechanical properties, corrosion resistance, and cost-effectiveness [26,27,28]. These mechanical properties are crucial for the durability and reliability of implants under the physiological stresses encountered within the human body [26,27,28]. Stainless steel screws, for instance, offer distinct torsional properties compared to titanium screws, providing surgeons with tactile feedback to prevent over-torqueing, a critical factor in the precise placement and longevity of these devices. Recent developments in nickel-free stainless steels are particularly noteworthy, not only for addressing nickel sensitivity concerns but also for their superior mechanical properties and improved corrosion resistance [26,27,28]. These new alloys represent a significant advancement in the materials science of biomedical implants, indicating a promising future for their application in more biocompatible and mechanically robust solutions.

#### 3.3.2. Surface Characteristics and Bioactivity

The surface characteristics of stainless steel implants are crucial for their performance in biomedical applications, including their integration into the body and resistance to corrosion. Stainless steel (SS), particularly AISI 316L, is extensively used in biomedical fields for making implants and medical devices due to its low cost, ample mechanical properties, corrosion resistance, and biocompatibility [28,29,30,31]. However, the success of these materials in biological environments hinges on their surface modification to enhance their biofunctionality, such as improving their resistance to fouling, preventing biofilm formation, and facilitating their integration with biological tissues [28,29,30,31]. Techniques used for these modifications include plasma vapor deposition, electrochemical treatment, and the attachment of different linkers that add functional groups to the surface. These modifications can significantly enhance the material’s corrosion resistance, osseointegration capabilities, hemocompatibility, and antibacterial properties [28,29,30,31]. For instance, sol–gel-derived coatings, such as nanostructured inorganic ZrTiO_4_ and hybrid ZrTiO_4_-PMMA thin films, have been shown to improve the biocompatibility of medical-grade stainless steel, correlating with enhanced surface characteristics [28,29,30,31]. Hydrophilicity and osteoconductivity are also key surface characteristics that can be tailored through surface modification. Modifying the surface to be more hydrophilic can enhance the implant’s interaction with biological tissues, thereby improving osseointegration and biocompatibility.

#### 3.3.3. Nanoscale Modifications and Biocompatibility

Modifying 316L stainless steel at the nanoscale, such as by adding selenium-coated nano-pit arrays, has been found to significantly boost both its compatibility with biological tissues and its ability to promote bone growth [27,32,33]. These enhancements not only elevate the material’s surface properties but also bolster its support for cellular attachment, growth, and bone-related activity [27,32,33]. Consequently, selenium-coated 316L stainless steel equipped with nano-pit arrays emerges as an excellent material for use in orthopedic and dental implants [27,32,33]. Another method for improving the longevity, bioactivity, and corrosion resistance of stainless steel involves the application of intense surface deformation processes like submerged friction stir processing [27,32,33]. This technique notably enhances the material’s resistance to bacteria and diminishes its corrosion rate by over 60%, while also reducing the release of harmful nickel and chromium ions [27,32,33]. The development of a stable, chromium oxide-rich passive layer and increased hydrophilicity play crucial roles in achieving these benefits. Moreover, the application of innovative nanocomposite coatings, such as natural hydroxyapatite/zircon (NHA/zircon), on stainless steel surfaces has been investigated to further improve their bioactivity and compatibility with biological tissues [27,32,33]. This strategy seeks to mimic the molecular structure of natural bone, enhancing the efficacy of dental and other biological implants.

### 3.4. Silicone

#### 3.4.1. Mechanical Properties

Recent advancements in silicone implant technology have enhanced their mechanical properties, such as deformation, resistance, and elasticity, contributing to their efficacy and safety [34,35,36]. This progress is critical for ensuring implants’ functional compatibility with the body’s tissues. Innovations in breast implant design aim to address common issues like capsular contracture by fostering a better understanding of the dynamic interactions between implants and surrounding tissues [34,35,36]. Tissue engineering, especially through the use of scaffolds, is paving the way for implants that more closely mimic natural breast tissue, promoting tissue regeneration and potentially reducing the reliance on non-degradable materials [34,35,36]. Silicone-based vascular prostheses, for example, have demonstrated stability and compliance comparable to those of traditional materials used in vascular surgery [34,35,36]. For customized silicone rubber-based implants, innovations in rapid curing processes have ensured the mechanical and biological integrity of these devices, enabling their use in applications ranging from cochlear implants to electrocortical grid arrays. 

#### 3.4.2. Surface Characteristics and Bioactivity

The surface characteristics of silicone implants play a pivotal role in their biomedical applications, impacting everything from their integration into the body to their interactions with biological tissues [37,38,39,40,41,42]. Silicone implants are widely utilized in the medical field for various applications due to their flexibility, biocompatibility, and durability. However, the success of these implants in the body is heavily dependent on enhancing their surface properties to improve their biocompatibility and functionality [37,38,39,40,41,42]. Techniques such as plasma surface modification and hydrophilicity adjustment have been explored to increase cell adhesion and proliferation on silicone surfaces. These modifications aim to create more bioactive surfaces that may limit biofilm formation and reduce the risk of implant rejection [37,38,39,40,41,42]. Plasma treatment, for instance, has been shown to significantly enhance the surface free energy of silicone, facilitating the stable attachment of functional coatings such as collagen, which, in turn, improves cell viability and adhesion [37,38,39,40,41,42]. This approach may enhance the biocompatibility of the implants whilst facilitating harmonious co-existence with surrounding breast and muscle tissue. Moreover, the modification of surface hydrophilicity through techniques like hydrogen plasma treatment can greatly influence the interactions between implants and biological tissues [37,38,39,40,41,42]. By improving the hydrophilicity of materials such as silicon carbide-based nanowires, researchers have demonstrated enhanced osteoblast adhesion and proliferation, suggesting potential for applications in bone tissue engineering. These advancements highlight the critical role of surface characteristics in the development of medical devices, aiming to achieve optimal integration and functionality within the body.

#### 3.4.3. Nanoscale Modifications and Biocompatibility [43,44,45,46,47]

Advances in the surface modifications of silicone implants, particularly breast implants, have shown a significant impact on their biocompatibility and integration with the body [43,44,45,46,47]. These modifications aim to address complications such as capsular contracture and breast implant-associated anaplastic large-cell lymphoma (BIA-ALCL), which have been exclusively associated with textured implant surfaces [43,44,45,46,47]. Recent studies have focused on altering the surface topography and chemistry of these implants to improve protein adsorption and cell adhesion, which are critical factors for the implant’s success and integration. The historical evolution of silicone breast implants reveals a trajectory towards increasing biocompatibility and functionality [43,44,45,46,47], from the first generation of thick, smooth silicone elastomer shells filled with Silastic, to the latest (sixth-generation) implants featuring biocompatible shells and radio frequency identification technologies [43,44,45,46,47]. Each generation introduced improvements aimed at enhancing the natural feel, reducing complications, and improving the safety profile of the implants. Functional biocompatibility testing of silicone breast implants based on surface roughness has introduced a novel classification system, categorizing surfaces into macro-, micro-, meso-, and nanotextured types. This classification aids in understanding how different surface characteristics influence the body’s response to the implant [43,44,45,46,47]. Studies have shown that progressive increases in implant surface texturing foster increasing levels of inflammation and the risk of developing BIA-ALCL, but potentially less capsular contracture or device malposition [43,44,45,46,47]. In contrast, no cases of BIA-ALCL have been reported in patients with a known history of exclusively smooth-surfaced implants. The use of smooth implants, however, comes at the expense of higher rates of capsular contracture or unwanted shifting of breast implants over time. Nanotextured-surfaced breast implants may provide a favorable compromise, whereby the risks of developing BIA-ALCL, capsular contracture, and breast implant malposition are all mitigated. Because BIA-ALCL usually takes 8–10 years to develop and is very rare, it is unlikely that informative clinical data reporting the rates of these particular clinical complications will be reliably available for several years. It is clear, however, that careful modification of the breast implant’s surface can have a critical impact on clinical outcomes.

## 4. Malignant Tumors Associated with Orthopedic Surgical Devices

Orthopedic implants have been used for nearly 100 years to stabilize fractures and as components of arthroplasties. The first orthopedic implants were composed of stainless steel and were later refurbished with Stellite, an alloy of chromium and cobalt, or titanium [48]. Non-metal materials, such as polyethylene and acrylic cement, are commonly used in arthroplasty procedures as well. The aforementioned materials are the primary components of orthopedic implants today and are considered to be biocompatible.

Still, some of these materials have demonstrated carcinogenic potential in animal studies [49,50,51]. In humans, implant-related malignancies are a rare but well-established possible long-term complication. Since 1956, there have been at least 92 case reports of orthopedic implant-associated malignancies, including 67 sarcomas, 18 hematological malignancies, 2 endothelial hemangioendotheliomas, 1 malignant peripheral nerve sheath tumor, and 1 squamous-cell carcinoma [52,53,54,55,56,57,58,59,60,61,62,63,64,65,66,67,68,69,70,71,72,73,74,75,76,77,78,79,80,81,82,83,84,85,86,87,88,89,90,91,92,93,94]. The majority occurred at the site of previous hip arthroplasties, with other sites associated with knee arthroplasties, intramedullary nails (IMNs), fracture fixation plates, and staples. Though the material composition is inconsistently documented, there have been reports of malignancies associated with cobalt–chromium, titanium, and stainless steel implants, with or without cemented or polyethylene components [52,53,54,55,56,57,58,59,60,61,62,63,64,65,66,67,68,69,70,71,72,73,74,75,76,77,78,79,80,81,82,83,84,85,86,87,88,89,90,91,92,93,94].

### 4.1. Orthopedic Implant-Associated Sarcomas

Sarcomas are a heterogeneous group of malignancies that arise from a mesenchymal cell origin, representing 0.8% of all cancers in the United States [95]. Of 67 case reports of orthopedic implant-associated sarcomas, 48 occurred at the site of hip arthroplasties (Table 1) [47,48,49,50,51,52,53,54,55,56,57,58,59,60,61,62,63,64,65,66,67,68,69,70,71,72,73]. Most diagnoses occurred between the fifth and seventh decades of life, at an average of 59.4 years [47,48,49,50,51,52,53,54,55,56,57,58,59,60,61,62,63,64,65,66,67,68,69,70,71,72,73]. The time from initial implant placement to cancer diagnosis ranged from 6 months to 31 years, with an average of 9.3 years.

The most common presentation includes pain, stiffness, edema, and pathologic fracture (Table 2). Radiographically, the tumors are adjacent to the implanted device and demonstrate osteolysis, cortical irregularities, permeative bone changes, pathologic fractures, and new periosteal bone formation. Biopsy was required for diagnosis.

Osteosarcomas and undifferentiated pleomorphic sarcomas were the most common sarcoma subtypes, occurring in 19 and 25 patients, respectively [47,49,50,51,52,53,55,56,57,58,61,63,64,65,66,67,68,72]. Angiosarcomas were reported in 11 patients [47,48,62,69]. The remainder of the cases consisted of rarer sarcoma subtypes. The gross pathology of tumor specimens demonstrated both bone and soft tissue infiltration. The majority of osteosarcoma cases arose in the periprosthetic bone [52,54,55,57,63,65,69,71]. Metastatic spread commonly involved the lungs. 

Treatment of these sarcomas depends primarily on tumor location and patient goals. Aggressive surgical treatment typically involves wide local resection, amputation, or hemipelvectomy. Of the 26 cases that reported the treatment approach, 5 and 14 patients received adjuvant chemotherapy and radiotherapy, respectively. Prognosis was poor, with death occurring within 2 years in most reported cases. 

### 4.2. Orthopedic Implant-Associated Lymphomas

Malignant proliferation of either T or B cells, otherwise known as non-Hodgkin lymphoma (NHL), is the second most common malignancy associated with orthopedic implants. Though most NHLs are intranodal, up to one-third can be found at extranodal sites. The most common of these is diffuse large B-cell lymphoma (DLBCL), comprising approximately 35% of all NHLs. 

There have been 18 reported cases of implant-associated lymphomas, including 9 associated with hip arthroplasty and 5 with knee arthroplasty, a majority of which arose from the bone–implant interface (Table 1) [80,81,82,84,85,86,87,88,92,93,94,96,97,98,99,100,101,102]. The remainder were found near fracture fixation plates or IMNs. Patients diagnosed with implant-associated lymphomas are slightly older than those diagnosed with implant-associated sarcomas, at 64.4 years. The mean latency from implantation to diagnosis is 8 years, with a range of 1 to 32 years. 

Most patients present with pain and swelling, suggestive of implant failure (Table 2). Thirty percent present with signs mimicking an underlying prosthetic infection, and fewer with enlarging masses, fevers, malaise, and weight loss. Lymphadenopathy has not been observed. General laboratory values vary, although elevated erythrocyte sedimentation rate and C-reactive protein, leukocytosis, or pancytopenia may exist. Most lesions are evaluated with plain radiographs followed by computed tomography (CT) or magnetic resonance imaging (MRI). Soft tissue mass with osteolysis, cortical irregularities, or periosteal reaction is commonly demonstrated, and surgical biopsy is required for diagnosis.

Surgical pathology demonstrates diffuse large B-cell lymphoma in 85% of reported orthopedic implant NHL cases [80,81,82,84,85,86,87,88,92,93,94,96,97,98,99,100,101,102]. Common histopathological findings include large tumor cells with hyperchromatic nuclei, coarse chromatin, and prominent nucleoli that stain for combinations of CD19, CD20, CD79a, and PAX5 [84]. 

Positron emission tomography (PET) is utilized to determine staging and guide treatment. Treatment includes combinations of surgical resection, hardware removal, chemotherapy, and radiotherapy. Where long-term outcomes are reported, 5 of 14 patients succumbed to their disease, while 9 experienced disease-free survival [82,84,85,87,88,92,93,94,96,97,98,101].

### 4.3. Pathogenesis

The proposed factors contributing to the pathogenesis of orthopedic implant-associated malignancies include wear debris, chronic antigen stimulation, and carcinogenic effects of biomaterials. Since the 1950s, solid implants in animal models have demonstrated carcinogenic impact, as demonstrated by Kirkpatrick et al. [49] in an experimental rat model evaluating the carcinogenic potential of subcutaneous implants composed of silicone, titanium, nickel chromium, cobalt–chromium, polyethylene, polymethylmethacrylate, or aluminum oxide. Tumors were found in 25.8% of specimens 2 years after implantation, with undifferentiated pleomorphic sarcomas most common. Polyethylene was most commonly associated with tumors (35%), while titanium was the least (12%). Further research expanded to 22 prosthetic materials and revealed increased sarcoma and lymphoma rates with metal alloy implants containing cobalt, chromium, or nickel [50].

Soluble metal ions produced by implants over time may impact carcinogenicity. Doran et al. [51] found marked increases in the incidence of carcinogenic transformation in mouse fibroblasts when exposed to soluble cobalt, chromium, and nickel salts. This was not found with their particulate composition. Previous animal models corroborated the lack of carcinogenic potential of particulate metals [103,104]. Increased levels of metallic ions in synovial fluid and blood in patients with joint prostheses may promote local tissue carcinogenesis, inducing DNA damage and promoting lymphogenic proliferation [105,106,107,108]. 

Wear debris and corrosion may play a role in carcinogenesis through the activation of macrophages and monocytes and the release of pro-inflammatory cytokines. Chiba et al. [109] demonstrated increased production of IL-6, IL-1, and TNF-a when human monocytes were incubated with polyethylene. These cytokines induce a foreign body type of chronic inflammatory reaction, causing aseptic loosening and periprosthetic osteolysis [110,111,112]. Chronic immune activation via antigen stimulation may contribute to local malignant transformation. However, this does not represent the only carcinogenic process, and a large majority of lymphocytic reactions associated with metallic corrosion products are benign in nature.

Large-sample studies evaluating the relationship between orthopedic implants and malignancies are inconclusive at best. Gillespie et al. [113] followed 1358 patients for 10 years after total hip arthroplasty (THA) and found increased incidence of lymphatic and hematopoietic cancers compared to the general population. Visuri and Koskenvuo [114] reported similar results in their cohort study of 433 Finnish patients with THAs. However, this registry was later updated to include 24,636 patients, and it demonstrated no difference in cancer incidence between patients with or without hip prostheses [115]. Mathiesen et al. [116] and Signorello et al. [117] independently followed large cohorts of Swedish patients with THAs and found no difference in cancer incidence. It is also notable that, despite the release of metallic debris to adjacent soft tissues, reports of malignancies around spinal hardware are lacking. 

## 5. Malignancies Associated with Plastic Surgical Devices

Breast implants are the most implanted medical devices in plastic surgery. Nearly 450,000 patients underwent implant-based breast augmentation or reconstruction in the United States in 2022 [118]. Breast implants are composed of a smooth or textured-surface silicone elastomer outer shell filled with either saline or highly cohesive silicone gel. Gluteal implants are less commonly used, employed in approximately 1000 cases in the United States annually. Gluteal implants are composed of either a textured or smooth silicone outer shell filled with highly cohesive silicone gel. 

Though breast and gluteal implants are approved by the Federal Drug Administration (FDA), they are not without risk. They have come to the forefront of discussion due to the discovery of BIA-ALCL, a rare T-cell lymphoma that most often presents as a delayed seroma surrounding a textured breast implant. The first case of BIA-ALCL was described in 1996 [91,119,120]. Twenty years later, BIA-ALCL was recognized by the World Health Organization (WHO) as a separate category of malignancy. Since that time, the National Comprehensive Cancer Network (NCCN) has published guidelines for the diagnosis and treatment of BIA-ALCL, emphasizing early intervention and surgical treatment [121]. 

Aside from BIA-ALCL, rarer malignancies associated with plastic surgical implants have been described, including breast implant-associated squamous-cell carcinoma (BIA-SCC), breast implant-associated diffuse large B-cell lymphoma (BIA-DLBCL), and gluteal implant-associated anaplastic large-cell lymphoma (GIA-ALCL).

### 5.1. Breast Implant-Associated Anaplastic Large-Cell Lymphoma

BIA-ALCL is an anaplastic large-cell kinase-negative (ALK-ve) CD30-positive T-cell lymphoma that typically presents as a delayed seroma surrounding a textured breast implant. Some patients develop an intracapsular mass exclusively or in conjunction with delayed seroma. Symptoms include pain, lymphadenopathy, skin rash, and fevers [121]. As of March 2023, the FDA has received 1352 medical device reports (MDRs) of BIA-ALCL (Table 3). BIA-ALCL’s incidence in patients with a history of textured implants is described as ranging from as high as 1:355 patients to as low as 1:40,000 patients [122,123,124,125,126]. Age at diagnosis is also variable, ranging from 24 to 90 years old. In contrast, the time from implantation to diagnosis has a consistently prolonged latency, with an average of 7–10 years [83,121,126,127]. 

The NCCN has published a standardized diagnostic algorithm for patients with a history of breast implants presenting with a delayed swollen breast concerning for BIA-ALCL [121]. Patients are first assessed with ultrasonography, CT, or MRI to assess for fluid collections, masses, or lymphadenopathy. Periprosthetic fluid collections are aspirated with ultrasound guidance, and associated masses are biopsied. Diagnosis is established with cytology, immunohistochemistry (IHC), and flow cytometry.

Histopathological analysis demonstrates large, pleomorphic cells with anaplastic morphology (Table 4). Their cell nuclei are large, oval, or multilobulated, with dense chromatin [128]. “Hallmark” cells, which have horseshoe- or kidney-shaped nuclei, are also commonly described [129]. BIA-ALCL is CD30-positive and ALK-negative, though additional biomarkers are commonly employed for diagnosis [130].

Total capsulectomy with a margin of normal tissue, otherwise termed en bloc capsulectomy, is the primary treatment. Intraoperative findings include intracapsular seroma fluid with or without tan nodular masses [130]. Immunotherapy, chemotherapy, and radiotherapy may be indicated for advanced disease. To date, there have been 63 reported deaths from BIA-ALCL [83]. Later stages of presentation are associated with increased mortality and recurrence risk [131]. Patients in remission should be monitored for recurrence every 3 to 6 months for 2 years [121].

### 5.2. Gluteal Implant-Associated Anaplastic Large-Cell Lymphoma

Owing to infrequent use of gluteal implants, only two cases of GIA-ALCL have been reported. Shauly et al. [132] reported the first case in 2019—a 49-year-old female who presented with metastatic ALK-negative ALCL after undergoing bilateral textured gluteal implant placement one year prior. Imaging studies demonstrated fluid collections and enhancement of both gluteal implants, which was later determined to be the source of malignancy. This patient expired from GIA-ALCL. Mendes et al. [133] reported another case in the same year—a 63-year old female who presented with a delayed periprosthetic seroma 9 years after implantation of textured gluteal implants. She was diagnosed with CD30+ GIA-ALCL on pathology following total capsulectomy, resulting in disease remission.

### 5.3. Breast Implant-Associated Squamous-Cell Carcinoma

Breast implant-associated squamous-cell carcinoma (BIA-SCC) is a rare, aggressive malignancy that originates within the breast implant capsule. Since the first case was described in 1992, there have been 19 verified cases of BIA-SCC according to the American Society for Plastic Surgeons (ASPS), with more cases under review (Table 3) [134,135,136,137,138,139,140,141]. 

Presentation includes unilateral breast pain, erythema, and fluid collection [134,135,136,137,138,139,140,141]. Cases have been reported in saline and silicone implants with either textured or smooth surfaces, although device history is not uniformly described [134,135,136,137,138,139,140,141]. The average age at diagnosis is 56.1 years, and diagnosis occurs between 7 and 42 years after initial implantation. Five cases occurred after breast augmentation and eleven after breast reconstruction (three were not reported). Diagnostic evaluation follows the NCCN guidelines, as for BIA-ALCL. 

Seroma aspirates confirming BIA-SCC express epithelial carcinoma marker CK 5/6 and squamous-cell transcription factor p63 on IHC, and they contain squamous cells and keratin on flow cytometry (Table 4). Tissue biopsy is required to confirm diagnosis. PET-CT should be performed prior to surgical intervention to determine the disease extent [142]. Findings consistent with BIA-SCC include an ill-defined mass arising from the breast capsule, with or without invasion of the chest wall.

Intraoperative findings include fungating breast capsule masses, thickened breast capsules, keratin debris, and turbid seroma fluid (Table 4) [130]. Histology reveals invasive keratinized squamous-cell carcinoma or metaplasia with evidence of acute or chronic inflammation. The current surgical treatment recommendation is aggressive, including total capsulectomy and radical mastectomy, as incomplete resection is associated with increased mortality and recurrence rates [142]. The overall prognosis is poor, with mortality by 6 months in 43.8% of reported cases. There is no described longevity benefit of adjuvant chemotherapy or radiotherapy.

### 5.4. Breast Implant-Associated Diffuse Large B-Cell Lymphoma

While most lymphomas associated with breast implants have a T-cell origin, there have been reported cases of B-cell lymphomas as well [90,130,143,144,145,146,147,148,149,150,151,152,153,154,155,156]. Of the 36 reported cases of implant-associated B-cell lymphoma, breast implant-associated diffuse large B-cell lymphoma (BIA-DLBCL) is the most common, with 25 total reported cases, 21 of which were Epstein–Barr virus (EBV)-positive (Table 3) [90,130,143,144,145,146,147,148,149,150,151,152,153,154,155,156,157]. 

Most patients report breast pain, swelling, or a palpable mass. Few present with capsular contracture, night sweats, fevers, hepatosplenomegaly, and lymphadenopathy (Table 4). The age at diagnosis ranges from 34 to 83 years of age and develops 6 to 44 years after implantation. Like BIA-ALCL, this malignancy is described in association with textured silicone breast implants. Diagnostic evaluation follows the NCCN guidelines for the evaluation for BIA-ALCL.

Diagnosis is made by seroma aspiration and tissue biopsy. Though most cases are EBV-positive, BIA-DLBCL may also be identified by B-cell markers CD20, CD19, CD79a, PAX-5, and BCL-6 on IHC. Histopathology reveals foreign-body giant cell reactions, pleomorphic lymphoid cells that show prominent nucleoli, atypical nuclei with numerous mitotic figures, and heterogeneous chromatin patterns. On gross pathology, BIA-DLBCL exhibits tan, thickened implant capsules with a coarse inner lining. Total capsulectomy is deemed adequate in the treatment of BIA-DLBCL localized to the implant capsule, with no role for chemotherapy or radiotherapy.

### 5.5. Pathogenesis

The etiology of BIA-ALCL remains unknown. There is a consensus that chronic inflammation associated with a textured surface is a key factor [158]. Beyond that, however, no theory has been verified, with host genetics, mechanical stress, bacterial infection, and implant toxins all being investigated [159,160,161,162,163,164,165].

Unlike many permanent implantable medical devices, breast implants are placed in a clean-contaminated site [158]. The breast is a unique organ with a diverse community of microbiota, with considerable variation from one individual to the next. Early on, a putative mechanism for the development of BIA-ALCL implicated bacterial infection with the Gram-negative bacterium *Ralstonia pickettii* [159], not typically found in the breast, but a common water-borne organism known to contaminate 16S genomic sequencing experiments [166]. More recent research has failed to identify a link between bacterial infection and BIA-ALCL. Interestingly, at least five cases of BIA-ALCL have been reported in transgender females, with similar presentation to their cis-female counterparts [167]. The microbial and hormonal *milieu* of the breast parenchyma has not been characterized in trans-females with the detail needed to enable comparison with cis-females. Unfortunately, investigation of other purported etiologies, including the impact of breast implant surface tribology or the release of particulates or toxins, has yet to offer convincing alternative explanations [160,161,162,163]. 

To date, no known driver rearrangements have been reliably linked to the development of BIA-ALCL [164]. However, whole-exome sequencing (WES) and whole-genome sequencing (WGS) of patients with BIA-ALCL have demonstrated key genetic alterations and chromosomal abnormalities. On a molecular level, BIA-ALCL demonstrates upregulation of hypoxia signaling proteins, specifically carbonic anyhydrase-9 [168]. Activating mutations in the JAK-STAT pathway have also been shown in next-generation sequencing performed on patients with BIA-ALCL [169,170]. 

In addition to mutations in STAT3 and NRAS, key players in the JAK-STAT pathway, critically deleted 7MB regions at the 11q22.3 chromosome have also been noted in multiple patient samples [164,169,171]. The protein kinase Ataxia-Telangiectasia Mutated (ATM) gene located on the 11q22.3 chromosome is frequently deleted in cases of chronic lymphocytic leukemia (CLL) [172]. The ATM protein kinase is a member of the phosphoinositide 3 lipid kinase (PIKK) family, which is involved in the DNA damage response. Specifically, double-strand DNA breaks activate ATM, which, in turn, phosphorylates serine or threonine residues of proteins involved in checkpoint regulation, apoptosis, and DNA repair [173].

In a comprehensive genetic analysis analyzing genome-wide copy number aberrations (CNAs), researchers interestingly found loss of 20q13.13 in 66% of samples [171]. This is specific to BIA-ALCL when compared to other classes of ALCL. Additionally, the interleukin-6-JAK1-STAT3 pathway was deregulated in mutational analyses of BIA-ALCL [171]. A similar deregulation in this pro-inflammatory, oncogenic pathway is characteristic of many cancers, including liver, breast, colorectal, lung, and pancreatic cancers [174]. Treatments targeting key molecules in this pathway are a promising key future direction for this body of research. 

Additionally, cytokine profiles have been developed to further characterize BIA-ALCL. BIA-ALCL demonstrates a Th2-type cytokine expression, with higher levels of IL-10, IL-13, and eotaxin as compared to reactive breast-implant seromas [175]. Additionally, an IL-10/IL-6 ratio >0.104 is consistent with diagnosis of BIA-ALCL [175,176,177]. With this categorization, targeted immunotherapies based on cytokine expression may represent a treatment approach.

In BIA-SCC, the malignancy arises from the capsule itself, not from the native breast tissue. At this time, the etiology and origin of BIA-SCC remain unknown. Some authors have hypothesized that BIA-SCC arises from squamous metaplasia arising from the transfer of ductal epithelium during implantation [134,135,141]. Chronic inflammation may disrupt cell apoptosis signaling pathways and cause tissue metaplasia [178]. This is well established in the esophagus and stomach in the form of intestinal metaplasia in response to long-term acid exposure. During the transition from acute to chronic inflammation, large quantities of immunosuppressive cells and cytokines invade the microenvironment. These cellular mediators are known to promote oncogenesis at a molecular level, signaling various chronic inflammatory signaling pathways implicated in carcinogenesis. Subsequent downstream signaling events affect the well-known intracellular pathways JAK-STST, nuclear factor kappa beta (NFkB), K-RAS, and p53, each of which has been implicated in oncogene activation, DNA damage and protein damage, and the release of reactive oxidative species (ROS) when dysregulated. However, future investigation is required to determine the exact molecular underpinnings of BIA-SCC [179,180].

BIA-DLBCL is another rare entity of breast implant-associated malignancies, of which a large majority of cases are EBV-positive. EBV-positive DLBCL has been associated with immunosuppression and chronic inflammation, categorized by the WHO as diffuse large B-cell lymphoma associated with chronic inflammation (DLBCL-CI) [85]. Further evaluation of BIA-DLBCL’s gross pathology reveals that it is more similar to fibrin-associated DLBCL (FA-DLBCL), a clinically separate subtype of DLBCL-CI [181]. This form of lymphoma is much more indolent in nature and has been reported in association with fibrin debris of atrial myxomas, endovascular graft thrombi, and pseudocysts. However, not all cases of BIA-DLBCL are EBV-positive, and further investigation is required to elucidate its tumorigenesis.

## 6. Malignancies Associated with Dental Implant Devices

According to the American Dental Association, over 3 million dental implants are placed annually by dentists and oral surgeons [182]. The demand for osseointegrated dental implants is substantial, with a global implant market already worth USD 13 billion, and scrutiny over the safety and efficacy of these devices has increased. Dental implants are primary composed of titanium and its alloys, with a small minority composed of zirconia. Both materials are biocompatible and safe to use in implanted materials. Nonetheless, dental implants are continuously exposed to the oral microbiota, facilitating bacterial soft tissue infection, biofilm formation, and peri-implantitis. Peri-implantitis affects approximately 10% of all dental implants and presents as hypertrophy, erythema, ulceration, and/or loss of bone support [183]. Certain neoplastic conditions, especially oral squamous-cell carcinoma (OSCC), can present similarly to peri-implantitis, posing a challenge to diagnosis. Though no established neoplastic potential of dental implants exists, an increased number of case reports document OSCC in the vicinity of dental implants.

### 6.1. Dental Implant-Associated Oral Squamous-Cell Carcinoma

Oral squamous-cell carcinoma (OSCC) is the most common malignant epithelial neoplasm of the oral cavity, pharynx, and salivary glands, representing 90% of oral cancer diagnoses. Heavy tobacco and alcohol use are synergistic risk factors, increasing one’s risk of developing OSCC by 16-fold when used together. Other risk factors include genetic predisposition, poor oral hygiene, immunosuppression, human papilloma virus (HPV) infection, and chronic inflammation. Dental implants may contribute to chronic inflammation and, in turn, promote tumorigenesis. There have been 71 reported cases of primary dental implant-associated malignancies, 69 of which were OSCCs (Table 5) [184,185,186,187,188,189,190,191,192,193,194,195,196,197,198,199,200,201,202,203,204,205,206,207,208,209,210,211,212,213,214,215]. 

Of the implant-associated OSCCs, 24 were recurrent cancers following surgical resection with dental restoration. The average age at diagnosis was 67 years old, with an average of 4.7 years from time of initial implantation to diagnosis. Most of the cases (87%) occurred in the mandible, while 10% and 3% occurred in the maxilla and tongue, respectively. Tobacco use is overrepresented in patients with implant-associated OSCCs compared to the general population, with 21 of 69 reported cases occurring in smokers.

Many patients with implant-associated OSCCs present with features mimicking peri-implantitis, including pain, inflammation, exophytic masses, ulceration, granulation tissue, and gingival hyperplasia (Table 6). Thus, diagnosis is often delayed, with many patients initially treated for more common and less sinister pathologies using antibiotics, oral hygiene optimization, and observation. 

Biopsy is required to confirm this diagnosis. Histopathology reveals dysplastic stratified squamous epithelium containing malignant epithelial cells with eosinophilic cytoplasm, hyperchromatic nuclei, increased mitotic activity, individual cell keratinization, and intercellular bridging. Imaging with panoramic X-ray, CT, or MRI determines the disease extent, with many tumors destroying underlying bone with areas of ill-defined radiolucency. 

Of the 50 cases with documented treatment algorithms, surgical excision alone was the primary therapeutic option. Chemotherapy and radiotherapy were used in 8 and 12 cases, respectively, with no demonstrated improvement in disease-free survival. Follow-up has been reported for 47 patients, of whom 37 patients demonstrated no residual disease post-treatment, with 6 deaths. Follow-up was not reported for 22 patients.

### 6.2. Pathogenesis

Several authors have reported the theoretical malignant potential of dental implants [195,201,216,217]. Though deemed an inert metal, particulate debris from titanium induces genomic instability in human fibroblast cells, mediated through chromatid breaks [216]. Others have hypothesized that dental implants provide a pathway for cancer invasion into adjacent bone [195,201,217]. Additionally, local inflammation in the form of peri-implantitis can augment corrosion and promote local tissue destruction. The inflammatory environment increases the release of cytokine mediators, especially prostaglandins, IL-1, IL-6, and TNF-α [201]; this imbalances cellular proliferation and apoptosis, possibly promoting neoplasia.

Molecular analyses of oral OSCCs have previously demonstrated dysregulation of the NF-kB pathway in their carcinogenesis [218,219]. Specifically, NF-kB is constitutively activated due to the high concentration of inflammatory mediators in the microenvironment, which, in turn, upregulates the expression of IL-6, IL-8, CCL5, and CXCL10 [219]. This promotes further chronic inflammation and tumorigenesis. Repression of E-cadherins is also a byproduct of this pathway, resulting in tumor migration and eventual metastasis [220]. Thus, NF-kB inhibition represents a possible future treatment for these malignancies.

Though these pathogenic mechanisms seem convincing, associations between dental implants and malignancy have not been corroborated by any population-based studies. Over one-third of all cases of dental implant-associated OSCC are recurrences, and are therefore likely caused not by the implant but, rather, by incomplete treatment. Moreover, the calculated standardized incident ratio of oral cancer after dental implant placement is extremely low, at 0.00017 [199]. A causative relationship is unlikely, or at least extremely difficult to prove.

## 7. Other Implant-Associated Malignancies

The bulk of implant-associated malignancies have been reported in the dental, orthopedic, and plastic surgery literature. Case reports in other specialties exist, specifically urological, vascular, and cardiac surgery. Herein, we review these cases and comment on any associated pathogenesis.

### 7.1. Malignancies Associated with Urological Devices

According to epidemiological studies, chronic bladder irritation, infections, cyclophosphamide-induced cystitis, urinary obstruction, tobacco use, and non-steroidal anti-inflammatory drugs are independent risk factors for bladder cancer. These inflammatory mediators induce squamous metaplasia of the bladder epithelium. Chronic indwelling urinary catheters (CIDCs) act as a local inflammatory irritant, promoting chronic bacterial infections, increasing the incidence of neoplasia in patients with or without spinal cord injury. This is corroborated by several population-based studies. Despite this association, it is unclear whether the inciting event is the CIDC itself or the sequelae of long-term urinary dysfunction. 

Although patients requiring CIDCs are at increased risk of developing bladder cancer, these cancers rarely arise from the device itself. There have been 12 case reports describing squamous-cell carcinoma arising from CIDCs (Table 7) [221,222,223,224,225,226,227,228,229,230,231,232]. Nine cases described tumor extension into the bladder. The foundation of therapy was surgical resection with or without cystectomy. Patients with advanced disease were offered palliative radiation, with five cases resulting in death. 

### 7.2. Malignancies Associated with Cardiac Surgical Devices

Cardiac pacemakers are electronic devices inserted to regulate cardiac rhythm and correct conduction disorders. Most cardiac pacemakers are primarily composed of titanium, though some older models contained mercury zinc. From 1976 to 2023, there have been 26 reported cardiac pacemaker-associated malignancies (Table 7) [242]. These include near-equal numbers of adenocarcinomas, sarcomas, and lymphomas. The average age at diagnosis is 78 years, with a 6.6-year lag between implantation and diagnosis. Most patients present with a painless, enlarging mass in the vicinity of the pacemaker. Treatment algorithms and prognosis are tailored to the cancer diagnosis.

The pathogenesis of these malignancies is unknown, but authors have hypothesized roles for chronic inflammation, genetic predisposition, or metal toxicity. One theory proposes prolonged electrical stimulation as a possible carcinogenic trigger [233]. Electrical fields have potentiating effects on cancer cells’ migration in vitro and could promote tumorigenesis adjacent to cardiac devices [233]. Currently, there is insufficient evidence to establish a clear link between pacemaker pocket tumors and the device itself.

### 7.3. Malignancies Associated with Vascular Surgical Devices

There have been 28 angiosarcomas and 4 lymphomas related to vascular grafts reported (Table 7) [233,234,235,236,237,238,239,240,241]. Of the angiosarcomas, 16 were linked to polyester prostheses (Dacron) and 4 were associated with polytetrafluoroethylene grafts. The remainder were not reported. The time from graft placement to diagnosis averaged 6.7 years. The vascular grafts were for the abdominal aorta in 19, thoracic aorta in 7, and lower extremity in 2 cases. Most patients presented with pain and unintentional weight loss. Proper diagnosis with CT imaging is challenging, as sarcomatous lesions can be confused with graft infection, pseudoaneurysm, or atypical endoleak. Treatment consists of surgical resection and radiotherapy. Prognosis is poor, with most patients expiring within one year. 

It is hypothesized that foreign material within the intima results in turbulent flow and chronic injury, promoting an environment of rapid cell turnover and oncogene activation. Furthermore, animal studies show that polyester prostheses such as Dacron can induce sarcomas [243]. Further investigation is required to elucidate any possible relationship in human models.

## 8. Limitations of Review

This literature review primarily aims to summarize existing knowledge on malignancies related to implantable medical devices and the carcinogenic potential of implant materials. A significant limitation is that it does not capture all available evidence, as it is not structured as a systematic review. Therefore, while providing a comprehensive overview of the subject matter, this review may not include the entirety of research findings relevant to each section discussed. Despite this limitation, this review strives to present a valuable synthesis of the current state of knowledge regarding implant-associated malignancies and material risks, serving as a foundational resource for further investigation and clinical consideration.

## 9. Conclusions and Future Directions

Aside from BIA-ALCL, which is associated with implant texturization, a clear cause–effect relationship has not been established between any implanted device or alloy and a specific malignancy. A common theory for these malignancies attributes carcinogenesis to chronic inflammation surrounding the implanted device. Though the link between inflammation and neoplasia is well established, solely naming chronic inflammation as the driver of disease is insufficient. More research in the form of animal models, population-based studies, investigation of predisposing or acquired genomic perturbations, and biochemical analyses is required to further elucidate these enigmatic cancer phenomena. 

## Figures and Tables

**Table 1 ijms-25-04653-t001:** Epidemiology and location of orthopedic implant-associated malignancies.

Malignancy	Total	Age	Location					Time from Implant to Diagnosis
			Hip	Knee	IMN	Staples	Plate	
Sarcoma [47,48,49,50,51,52,53,54,55,56,57,58,59,60,61,62,63,64,65,66,67,68,69,70,71,72,73]	70	59.6 *	51	2	7	3	2	9.4 *
Osteosarcoma	19							
Undifferentiated Pleomorphic Sarcoma	25							
Angiosarcoma [47,48,62,69]	11							
Other	16							
Lymphoma [75,77,81,84,85,87,88,89,91,92,93,94,95,96,97]	18	64.4 *	9	5	1	0	3	8.0 *
DLBCL	15							
Other	3							

* Represents average.

**Table 2 ijms-25-04653-t002:** Clinical characteristics of orthopedic implant-associated malignancies.

Malignancy	Presentation	Radiographic Findings	Pathology/Histology	Intraoperative Findings	Treatment	Prognosis
Sarcoma [44,45,46,47,48,49,50,51,52,53,54,55,56,57,58,59,60,61,62,63,64,65,66,67,68,69,70,71,72,73]	Pain Soft tissue mass Edema Pathologic fracture	Osteolytic destruction Cortical irregularities Permeative bone formation Periosteal reaction Lung metastases	Malignant stromal cells Abundant atypical mitoses Nuclear pleomorphism Multinucleated giant cells	Tan-grey or tan-yellow mass Hemorrhagic Necrosis Cystic components	Resection/Amputation (34) Chemotherapy (13) Radiotherapy (17) No treatment (7) Not reported (16)	Death (37) Remission (10) Not reported (23)
Lymphoma [80,81,82,83,84,85,86,87,88,89,90,91,92,93,94,95,96,97,98,99,100,101,102]	Pain Soft tissue mass Cutaneous nodules Chronic draining sinus Fevers/chills	Osteolytic destruction Cortical irregularities Periosteal reaction	Hyperchromatic, enlarged lymphocytes Prominent nucleoli CD20+	Granulomatous appearance Mimics infection White gelatinous material	Radiotherapy (12) Chemotherapy (10) Resection/Amputation (3) Not reported (2)	Death (5) Remission (9) Not reported (5)

**Table 3 ijms-25-04653-t003:** Epidemiology of plastic surgery implant-associated malignancies.

Malignancy	Total Cases	Age	Time from Implant to Diagnosis
BIA-ALCL [91,118,119,120,121,122,123,124,125,126,127,128,129,130]	1352	54 **	10 **
BIA-SCC [130,131,132,133,134,135,136,137]	19	56.1 *	20.7 *
BIA-DLBCL [86,125,138,139,140,141,142,143,144,145,146,147,148,149,150,151,152]	25	60.5 *	14.1 *

* Mean. ** Median.

**Table 4 ijms-25-04653-t004:** Clinical characteristics of breast implant-associated malignancies.

Malignancy	Presentation	Radiographic Findings	Pathology	Intraoperative Findings	Treatment	Prognosis
BIA-ALCL [91,118,119,120,121,122,123,124,125,126,127,128,129,130]	Pain Unilateral swelling	Fluid collection around implant Intracapsular mass	Pleomorphic anaplastic cells Prominent nucleoli Dense chromatin Hallmark cells CD30+, ALK-	Intracapsular turbid, viscous fluid collection Speckling of inner surface of capsule	Total capsulectomy, implant removal, extracapsular mass resection Chemotherapy and radiotherapy for late stages	2.8% 1-yr mortality 5% 5-yr mortality
BIA-SCC [130,134,135,136,137,138,139,140,141]	Pain Unilateral swelling Capsular contracture	Fluid collection around implant Posterior capsular mass	Invasive SCC/metaplasia Keratin debris CK5+, CK6+, p63+	Fungated mass Keratin debris Tan/yellow capsules Viscous, turbid fluid collection	Total capsulectomy, implant removal, radical mastectomy No role for chemotherapy or radiotherapy	43.8% 6-mo mortality
BIA-DLBCL [130,147,148,149,150,151,152,153,154,155,156,157]	Pain Unilateral swelling Fevers, night sweats	Fluid collection around implant	Giant cell reaction Pleomorphic lymphoid cells Atypical nuclei CD20+, CD19+ 𝛋- or ƛ- light chain restriction	Tan thick capsules Gritty inner capsular lining Necrotic fibrinous material	Total capsulectomy, implant removal, extracapsular mass resection No role for chemotherapy or radiotherapy	No deaths recorded

**Table 5 ijms-25-04653-t005:** Epidemiology and location of dental implant-associated malignancies.

Malignancy	Total	Age	Time From Implant to Diagnosis	Location			Tobacco/Alcohol Use	Recurrence/Metastasis
				Mandible	Maxilla	Tongue		
OSCC [184,185,186,187,188,189,190,191,192,193,194,195,196,197,198,199,200,201,202,203,204,205,206,207,208,209,210,211,212,213,214]	69	66.9 *	4.7	60	7	2	21	24
Plasmocytoma [209,215]	2	75.0	20.1 *	2	0	0	NR	1

* Represents average. NR—not reported.

**Table 6 ijms-25-04653-t006:** Clinical characteristics of dental implant-associated malignancies.

Malignancy	Presentation	Radiographic Findings	Pathology/Histology	Treatment	Prognosis
OSCC [115,116,117,118,119,120,121,122,123,124,125,126,127,128,129,130,131,132,133,134,135,136,137,138,139,140,141,142,143,144,145,146,147,148,149,150,151,152,153,154,155,156,157,158,159,160,161,162,163,164,165,166,167,168,169,170,171,172,173,174,175,176,177,178,179,180,181,182,183,184,185,186,187,188,189,190,191,192,193,194,195,196,197,198,199,200,201,202,203,204,205,206,207,208,209,210,211,212,213,214]	Exophytic mass Ulceration Inflammation Verrucous lesion Leukoplakia Gingival hyperplasia Erythema Granulation tissue	Osteolytic destruction Irregular soft tissue mass	Hyperkeratosis Dysplastic squamous cells p53+	Surgical excision (50) Chemotherapy (8) Radiotherapy (12) Not reported (19)	Death (6) Remission (37) Not reported (22)

**Table 7 ijms-25-04653-t007:** Other implant-associated malignancies.

Surgical Field	Malignancy	Total
Cardiac Implants [223]	Adenocarcinoma	8
	Lymphoma	6
	Carcinoma	6
	Other	6
Urologic CICDs [221,222,223,224,225,226,227,228,229,230,231,232]	Squamous Cell Carcinoma	12
Vascular Grafts [233,234,235,236,237,238,239,240,241]	Angiosarcoma	15
	Lymphoma	4

## Data Availability

No new data were created or analyzed in this study. Data sharing is not applicable to this article. All information relevant to this systematic review is part of the manuscript, figures and tables. If any further information is required, the reader may contact the corresponding author for clarifications.

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
