# Peer review of "An Update on Implant-Associated Malignancies and Their Biocompatibility"

_ijms, 2024, doi:10.3390/ijms25094653_

Round 1

Reviewer 1 Report

Comments and Suggestions for Authors

This paper deals with the question to what extent different implants have a carcinogenic potency. Various medical implants are included in the assessment and their probability of malignant transformation is presented.  This work is very comprehensive and very detailed and provides an interesting overview. 

Reviewer 2 Report

Comments and Suggestions for Authors

The Authors present a comprehensive review of malignant tumors associated to various types of implants. The following comments and suggestions aim at improving the manuscript and readership to a wider audience.

Line 105. 2.2 Cobalt-Chromium. The alloy has a percentage of Mo as correctly reported in 2.2.3 Nanoscale Modifications and Biocompatibility. It is suggested to modify in Cobalt-Chromium-Molybdenum.

Line 124. Surface Characteristics and Bioactivity. It is unclear how implant components made of CoCrMo alloy in orthopedic implants are involved in osseointegration since those components are formed of Ti alloy. Please specify which kind of current implants use CoCrMo at bone interface.

Line 156. Osseointegration instead of osteointegration.

Line 243. It is unclear how these modifications can promote "better integration" with "bodily tissues". For example, silicon implants are used for small joint, digit prosthesis and the aim is of a minimal synovial reaction. In breast implants, fibrous reaction should be minimized too and it is suggested to explain more clearly the sequence of events for this kind of improvement.

Line 257. Similar to the above comment. How the modifications can reduce complications such as ALCL if it is unkown how this complication is developed?

Line 280. It is a new subchapter describing the occurrence of malignant tumors. It is suggested to modify the title as 3.0 Malignant tumors associated to orthopedic surgical devices.

Line 290. 2 "endotheliomas" should be epithelioid hemangioendotheliomas or angiosarcomas, "peripheral nerve tumor" should be malignant peripheral nerve sheath tumor, "epidermoid cancer" should be better defined.

Line 312. Osteosarcomas in peripheral limbs usually occur during the first two decades of life with a late age peak predominantly associated with Paget's disease. It would be interesting to add some more information regarding the hstological subtypes of these OS and regarding pleomorphic sarcomas if originated by surrounding soft tissues with bone involvement or diectly from periprosthetic bone. It would be interesting also to know if there have been recent reports or if the majority is confined to old literature.

Line 322. Table 1 has no title. 

Line 325.  Regarding malignant lymphomas, it is unclear if they originated from the periprosthetic bone or the periprosthetic soft tissue. In the latter instance, some comments should be made regarding the recent occurrence of florid lymphocytic reactions associated with metallic corrosion products and non malignant transformation reported so far even with formation of reactive germinal centers. 

Line 368. Some comments might be added to nanoparticles toxicity/carcinogenic potential. See above comment regarding the "lymphogenic proliferation" of obscure significance. Care should be exercised in citing one source of experimental studies. Thousands of patients have experienced macrophage induced PE osteolysis; however, no cases of bone sarcomas have been reported to the best of knowledge of this reviewer. 

Line 380. Is there any clinical evidence that "Chronic immune activation via antigen stimulation may contribute to local malignant transformation"? It is a generic sentence without any details and it is suggested to delete it. 

Line 382. Population-based studies for such rare occurrences need much larger samples of population to be reliable in the order of at least hundreds of thousands if not millions. The studies cited are too small to support the statement of "mixed results" which are at best inconclusive. 

No cases of malignancies are reported in this review around spinal metal devices, in spite of evidence of abundant metal release into the adjacent soft tissues. The Authors should provide a comment regarding this discrepancy.

Line 414. BIA-ALCL is an interesting entity because of its unusual development within the peri-implant fibrous capsule without evidence of a florid lymphocytic reaction and resolved in most of the cases by breast implant and capsule removal without adjuvant therapy. A unique occurrence not replicated in any case of orthopedic implant periprosthetic pseudocapsule. The authors should expand in the Pathogenesis subchapter regarding this lack of evidence for chronic inflammatory reaction compared to the orthopedic implants and the relation with the material in a very small subset of patients by default all females (the occurrence in male-to-female transgender population with breast augmentation has not been reported to the best of my knowledge). 

Line 542. Is the BIA-SCC secondary to squamous cell metaplasia? and if so, of breast duct after implant? Is there any histological evidence described or only a non-verified hypothesis? Breast duct carcinomas may have a squamous cell component but SCC from squamous metaplasia should be similar to the one of the respiratory epithelium in smokers more than the example of Intestinal metaplasia-adenocarcinoma at the gastro-esophageal junction. However, in this case there would be no chemical toxicity/carcinogenicity to justify it. 

Line 598. Table 5. Were the 2 cases of plasmocytoma confined to dental peri-implant location?

Line 609. Table 6. Cases of advanced SCC wil show associated chronic inflammation with or without dental implant association due to ulceration of the oral mucosa. Therefore, the findings are non-specific of peri-implantitis.

Line 618. "Several authors..." is missing citations. Anything might have a theoretical potential but at least some evidence should be shown to confirm it. 

Line 666. Which kind of adenocarcinomas would be associated with cardiac pacemakers? Tissue of origin should be specified. Please provide citation for the theory of electrical stimulation from a pacemaker as potential carcinogenic trigger. 

Line 698. Conclusions. "In the past 10 years, reports of implanted device-related malignancies have drastically increased". Are only the reports increased or the prevalence of the occurrence increased based on the number of operations performed? The conclusions should address that up to the present time, no specific type of malignant tumor has been associated to any specific material alloy or device. Moreover, in vitro and/or animal "models" would not be adequate to predict behavior in humans in pre-clinical testing and most probably useless, especially the former. 

Reviewer 3 Report

Comments and Suggestions for Authors

This comprehensive review investigates the epidemiology, clinical presentations, pathology, and mechanisms of carcinogenesis associated with prosthetic devices. It delves into the impact of surgical factors and implant properties on the development of malignancies. While intriguing, several issues require attention:

- The clarity of the tables is lacking. As this is a review, it's imperative to cite the sources of the data presented and specify the context of the numbers

- PROSPERO guidelines haven't been utilized, which could enhance the rigor of the study

- The legend for Table 1 appears distant from the table itself

- The article selection and exclusion criteria are absent, necessitating improvements in the methodology section

Reviewer 4 Report

Comments and Suggestions for Authors

Dear Authors,

I have read your paper "An Update on Implant-Associated Malignancies" carefully.

 Explanations are clear and the review is easy to read.

 However, it requires few corrections

 Has the PRISMA guidelines been followed?

Please add, the methodological sections in which it is described in detail how the review has been done.

How can the reader know if the review is a complete review?

The sections content includes group citations of articles. Especially in a review article, such groupings should be avoided. More details on the quoted papers should be provided.

The paper can be accepted for publication after major improvements.

Round 2

Reviewer 4 Report

Comments and Suggestions for Authors

Dear Authors,

I have read your modified paper "An Update on Implant-Associated Malignancies" carefully.

 Now, explanations are clear, and the review is easy to read.

 The paper can be accepted for publication.